# Adapting spatiotemporal gait symmetry to functional electrical stimulation during treadmill walking

Seung-Jae Kim [ID]*, Alex Worthy, Brandom Lee, Setareh Jafari, Olivia Dyke, Jeonghee Cho, Elijah Brown

Biomedical Engineering, California Baptist University, Riverside, CA, United States of America

* sjkim@calbaptist.edu

## Abstract

Individuals with neurological impairments often exhibit asymmetrical gait patterns. This study explored the potential of using functional electrical stimulation (FES) as a perturbation method during treadmill walking to promote gait symmetry adaptation by investigating whether the FES perturbation could induce gait adaptation concerning spatial and temporal gait symmetry in healthy subjects. In the FES perturbation, both legs received electrical pulses at the same period as the subjects' initial stride duration, and the temporal gap between the two pulses for each leg was manipulated over a 7-min period. Following this, subjects continued to walk for another 5 minutes without FES. Subjects participated in two trials: implicit and explicit. In the implicit trial, they walked comfortably during FES perturbation without consciously adjusting their gait. In the explicit trial, they voluntarily synchronized their toe-off phase to the stimulation timing. To examine the effects of the FES perturbation, we measured step length and stance time and then analyzed changes in step length and stance time symmetries alongside their subsequent aftereffects. During the explicit trial, subjects adapted their gait patterns to the electrical pulses, resulting in a directional change in stance time (temporal) symmetry, with the left stance becoming shorter than the right. The stance time asymmetry induced by FES perturbation showed a slight residual effect. In the implicit trial, the directional change trend was slightly observed but not statistically significant. No consistent trend in step length (spatial) symmetry changes was observed in either condition, indicating that subjects may adapt their spatial gait patterns independently of their temporal patterns. Our findings suggest that the applied FES perturbation strategy under explicit condition can induce adaptations in subjects' temporal gait asymmetry, particularly in stance. The implicit condition showed a similar slight trend but was not statistically significant. Further experiments would provide deeper understanding into the mechanism behind subjects' response to FES perturbations, as well as the long-term effects of these perturbations on the spatial and temporal aspects of gait symmetry.

**Data Availability Statement:** "data used for analysis" file is available from the google database at the following link: Kim, Seung-Jae (2024). Adapting spatiotemporal gait symmetry to electrical stimulation during treadmill walking

[Dataset]. Dryad. https://doi.org/10.5061/dryad.
d7wm37q7w.

**Funding:** The author(s) received no specific
funding for this work.

**Competing interests:** The authors have declared
that no competing interests exist.

## Introduction

Gait asymmetry frequently follows neurological impairments such as stroke, Parkinson's disease (PD), or multiple sclerosis (MS) [1, 2]. The achievement of gait symmetry is an important goal in rehabilitation, as it can significantly alleviate issues experienced by many neurologic populations. These issues include reductions in several areas such as walking speed, stability-balance [3, 4], general activity levels, and load-related aspects potentially affecting bone density [5]. Perturbation-based gait training strategies have emerged as effective methods for enhancing and treating gait asymmetry. Rehabilitation robots, for instance, can be integrated into exoskeletons to guide or perturb the patient's walking patterns, promoting efficient adaptation and correction [6–8]. The split-belt treadmill (SBT), a potent rehabilitation tool, effectively alters patients' gait patterns through short-term motor adaptation [9, 10], leading to aftereffects following adaptive training [11, 12]. Typically, when subjected to external perturbations during motor training, the nervous system reacts by trying to reduce errors from one movement to the next, showcasing its capacity for rapid adjustment. This critical attribute can be effectively utilized within a rehabilitation program [13].

An alternative approach, quite distinct from utilizing mechanical devices for administering perturbations, involves functional electrical stimulation (FES). FES is a well-established technique used in gait rehabilitation; individuals often pair FES with treadmill or overground walking to activate critical muscles involved in walking or facilitate the reorganization of existing neural circuits by timing the stimulation in sync with the different gait phases [14, 15]. While FES can bring about many other effects on gait rehabilitation, in this study, we explored a novel way to utilize FES to perturb gait to induce gait symmetry adaptation. In our previous study [16], we demonstrated that the recurring sensorimotor stimulation provided by FES during treadmill walking could contribute to implicit changes in the gait cycle period: healthy subjects tended to spontaneously synchronize their gait phase with the timing of brief electrical pulses. Consequently, these synchronized responses resulted in alterations in the gait cycle period, corresponding to changes in pulse frequency.

We extended this idea and developed a hypothesis that skillfully manipulating the timing of electrical pulses applied to both legs could induce asymmetric gait patterns aligned with the gait characteristics synchronized with the external stimulations. To test this hypothesis, we devised an FES-based perturbation paradigm in which the period of electrical pulses applied to both legs is the same, but the time intervals of the two pulses can be adjusted over time: the adjusted time interval of the two pulses within a gait period was regarded as the perturbation level in this study. As the first objective of this study, we investigated whether such a perturbation paradigm utilizing FES could induce changes in spatial and temporal gait patterns while subjects voluntarily synchronized a specific gait phase of both legs with the perturbations (explicit condition). Motor learning includes a form of learning process known as use-dependent plasticity, which involves developing movement biases that favor repeated movement directions [17]. Thus, we hypothesized that the FES perturbation could induce asymmetric gait patterns through this learning process. Meanwhile, motor learning is a result of the combination of explicit (voluntary) and implicit (involuntary) adaptation processes [18]. Implicit adaptation has garnered significant attention in motor rehabilitation due to its potential for enhancing the retention of learned motor patterns from training [19–21]. Therefore, we also investigated whether the FES perturbation could lead to implicit changes in asymmetric gait patterns. To explore this, we also conducted implicit trials where subjects walked comfortably without attempting to compensate for their gait phase with the electrical stimulation. We hypothesized that the FES perturbation could, to some extent, induce asymmetric gait patterns depending on the degree to which an individual spontaneously adjusts their gait in response to

FES perturbations. In this context, our study would support the idea that inducing asymmetric gait patterns could lead to gait symmetry among individuals who initially present with asymmetric gait. Therefore, a positive outcome of this study would imply that gait asymmetry adaptation through an FES perturbation paradigm holds promise as a potential method for gait asymmetry rehabilitation programs.

## Methods

### A. Subjects

Eighteen subjects (21.0 ± 2.8 years) participated in this study. Before obtaining their written informed consent, we provided the subjects with a detailed explanation of the study procedure. Subjects with an irregularity in their walking pattern, such as bow-leggedness, were excluded as this may interfere with our motion sensor's ability to capture data. The subjects were informed of the physical fatigue they would experience during the experiment and were accustomed to walking on a treadmill. All protocols (065-1617-EXP) were approved by the Institutional Review Board of California Baptist University to ensure that the study adhered to national and international guidelines for research on humans. Additionally, all the researchers gained CITI certifications in Biomedical Researchers and Responsible conduct of research as an additional safeguard in adhering to regulations and protocols of working with subjects in a research environment. The recruitment phase for this study commenced on 02/05/2023 and ended on 08/15/2023.

### B. Experimental setup

All experiments were performed on a treadmill (Woodway USA, Waukesha, WI) equipped with supporting handrails to ensure the safety of subjects. All subjects were given an approximate 5-minute preparatory period to adjust to walking on the treadmill. Two motion capturing markers were attached to the back of the subjects' shoes and tracked by a camera-based motion-capturing system (Optotrak 3D Investigator, Norther Digital Inc. Canada). This system located the positions of the markers and sent the data to a PC in real-time using a program developed with LabVIEW (National Instruments Corp., TX) to measure the subjects' spatio-temporal gait parameters such as step length, stride period, and stance duration. In this study, the position of the most forward marker was designated as the heel strike position, while the most backward marker position was identified as the toe-off position. We determined the stride period by measuring the interval from one heel-strike to the next, and the stance duration by measuring the interval from a heel-strike to the subsequent toe-off. It is important to note that the actual instance of toe-off occurs slightly after the marker reaches its most backward position. Nonetheless, we adhered to these measurement criteria consistently, prioritizing the assessment of symmetry between the two legs.

In all trials, subjects walked on a treadmill while being hooked up to a commercial electrical stimulator (MOTIONSTIM8, Krauth+Timermann, German), controllable through a computer program. Two 2" x 2" square electrodes, placed approximately 10 cm apart on the calf and the lower ankle, were used to trigger ankle movement by contracting the calf muscle (Fig 1A). Stimulation locations for the calf group included the sural nerve, gastrocnemius, peroneus, or soleus. The electrical stimulator was battery-powered, and a current isolator was used in conjunction with it for added safety. The stimulation threshold of each subject was determined by the magnitude of the current that induced slightly noticeable ankle plantar flexion. The preferred amplitude ranged from 20 mA– 30 mA during the trials. After documenting subjects' preferred stimulation amplitude and walking speed, an experimental trial ensued where their preferred treadmill speed and amplitude threshold were employed.

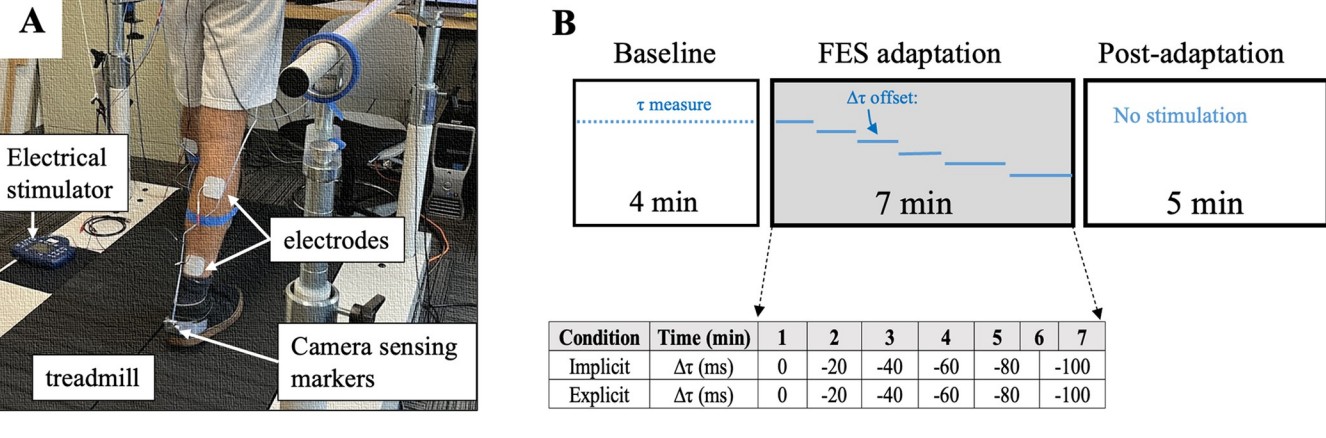

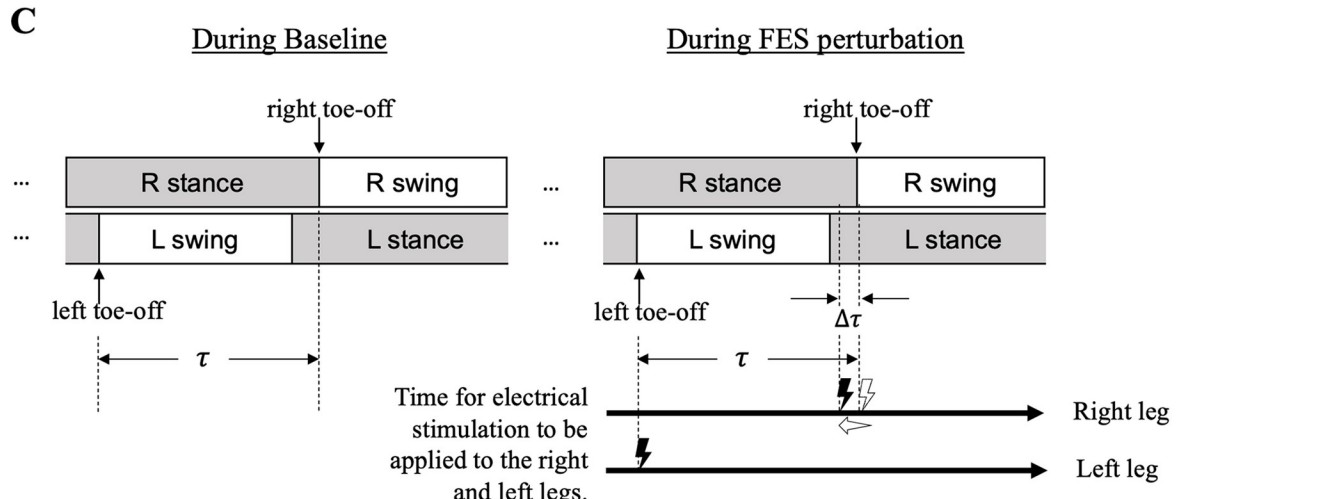

**Fig 1.** (**A**) Experimental setup showing a subject's trial with electrodes placed on the lower legs. (**B**) Each 16-minute trial consisted of a 4-minute baseline, a 7-minute perturbation, and a 5-minute post-perturbation period. For the perturbation period, the offset time was gradually adjusted to 0, -20, -40, -60, -80, and -100 ms from the initial τ. (**C**) During the initial FES perturbation period, electrical pulses were delivered to both legs at the same frequency as the initial stride, but the timings of the two pulses were separated by time τ, which signifies the initial time separation between the left toe-off and the subsequent right toe-off. Gradually, the timing interval between the two pulses was decreased by Δτ.

## C. Experimental protocol

Experiments were conducted with the following objectives: a) to primarily investigate the impacts of the FES perturbation on spatiotemporal gait symmetry among healthy subjects and b) to explore how the effects varied between the subjects' explicit and implicit actions during training. Accordingly, all subjects were required to participate in two distinct trials separated by several days: a trial with implicit condition and a trial with explicit condition. The implicit trials were always completed prior to the explicit trials. For the implicit trial, which involved unconscious reaction to FES, subjects were instructed to walk normally and comfortably through the trial. The subjects received no specific instructions, aside from being informed about the periodic electrical stimulation that would occur during the trials. Conversely, the explicit trial required deliberate adjustments, with subjects consciously entraining their gait to the timing of electrical stimulation. Specifically, the subjects were asked to explicitly synchronize the toe-off phase of their gait for both legs with the stimulation.

Each 16-minute trial consisted of three stages: a 4-min baseline period, a 7-min perturbation period, and a 5-min post-perturbation period (Fig 1B). During the baseline period, we measured and calculated the average time ($\tau$) between the left-toe-off and the subsequent right-toe-off phases, as well as the average stride duration from each gait cycle. During the perturbation period, we administered electrical pulses (0.12-second duration, 200 μs biphasic pulses at 90 Hz, 20–25 mA intensity) to both the right and left legs, at the same period of the baseline stride duration. In other words, periodic electrical stimulation was applied to each leg independently, but the stimulations were not delivered simultaneously to both legs; there was a time interval of $\tau$ between the stimulations. The time interval between the two electrical pulses delivered to both legs was gradually decreased by the offset time $\varDelta\tau$ (Fig 1C). During the initial 60 seconds of the perturbation period, the electrical pulses were applied with zero offset to acclimate subjects to the stimulation. After that, the offset time ($\Delta\tau$) was gradually adjusted to -20 ms, -40 ms, and -60 ms for 60 seconds each, and -80 ms and -100 ms for 90 seconds, respectively (a 100 ms offset time was approximately 7.5% of the average gait cycle duration in this study). In essence, the separation time between two pulses delivered to both the right and left legs was reduced by $\Delta\tau$. This manipulation was intended to make the right-toe-off phase occur slightly earlier than in the typical gait cycle, thus inducing gait asymmetry during the perturbation period. Following the perturbation period, electrical stimulation was stopped, and subjects continued to walk for the remaining 5 minutes of the post-perturbation period.

## D. Data analysis

The primary measures analyzed in this study were step length and stance duration time. The step length was determined as the distance between the marker on each leg at the heel strike of the leading leg. Specifically, right step length refers to the distance measured at the right heel strike, while left step length refers to the distance measured at the left heel strike. The gait cycle typically consists of two phases: stance and swing. The stance duration was calculated as the percentage of time the foot remained in contact with the ground during the gait cycle (stride time). Right stance duration represents the percentage of time the right foot was in contact with the ground, while left stance duration represents the percentage of time the left foot was in contact with the ground. To evaluate the symmetry of step length and stance duration, a ratio (%) between the measurements of the right and left legs was computed for each gait cycle by using the following formula: 100×(right leg measure–left leg measure)/(0.5×(right leg measure + left leg measure)). For example, a positive step symmetry ratio indicates that the right step length is longer than the left step length, while a negative symmetry ratio indicates that the right step length is shorter than the left step length. In addition, double limb support time is defined by the time in which both feet are in contact with the ground. There are two periods of double support per gait cycle: right double support and left double support. We define *right double support* as occurring at the end of the right limb's stance (i.e., the time from left heel strike to right toe-off) and *left double support* at the end of the left limb's stance (i.e., the time from right heel strike contact to left toe-off).

For data analysis, the mean and standard deviation (SD) of the symmetry ratios were calculated in successive 30-second intervals to examine the changes in symmetry during the perturbation and post-perturbation periods. The group means and SDs of the step length symmetry and stance duration symmetry were then calculated across all the subjects. To minimize between-subject variability in gait symmetry, the baseline symmetry was used as a reference. The baseline symmetry, which represented the symmetry value measured from the baseline period, was subtracted from the measured symmetry data obtained during all perturbation and post-perturbation periods (including the baseline itself) for each subject.

The primary objective of this study was to investigate the presence of spatiotemporal gait asymmetry induced by the FES perturbation during treadmill walking. We conducted statistical analysis for both the 7-minute perturbation period and the subsequent 5-minute post-perturbation period separately. Paired t-tests were first utilized to determine significant changes in step length and stance duration symmetries, compared to the baseline. These tests allowed us to identify the specific perturbation levels (the offset time) that exhibited significant changes in symmetry. Notably, each perturbation level in our trials lasted either 60 or 90 seconds, and we used the symmetry value obtained from the last 30-second window of each perturbation level. For the post-perturbation period, we assessed only three 30-second epochs (early post-perturbation, middle post-perturbation, late post-perturbation). Additionally, a one-way repeated measures ANOVA was performed to determine how increasing perturbation levels affected changes in step length and stance duration symmetries. All statistical tests were performed at a significance level of $p \leq 0.05$, and the p-values are presented in the main text unless depicted in the figures.

## Results

Fig 2A illustrates an example of changes in step length and stance duration over time obtained from a subject during the explicit trial. Each data point represents a measurement from each gait cycle. The changes in group mean step length and group mean stance duration across various epochs (30-second intervals) during the explicit trial are shown in Fig 2B. Throughout the perturbation period, as the perturbation level (indicated by the time separation between two electrical pulses sent to the right and left legs) gradually changed, we noticed a slight elongation in the right step length compared to the left. Additionally, the stance duration for the left side decreased compared to the right side. When the stimulation stopped, the relative difference in step length and stance duration changes between the right and left legs appeared to revert to normal, resembling the patterns observed during the baseline period.

In the explicit condition, subjects exhibited a prolonged stance duration on their right leg (or a reduced stance duration on their left side) throughout the perturbation period, leading to a positive asymmetry ratio (Fig 3B). Paired t-test results revealed that the stance symmetries

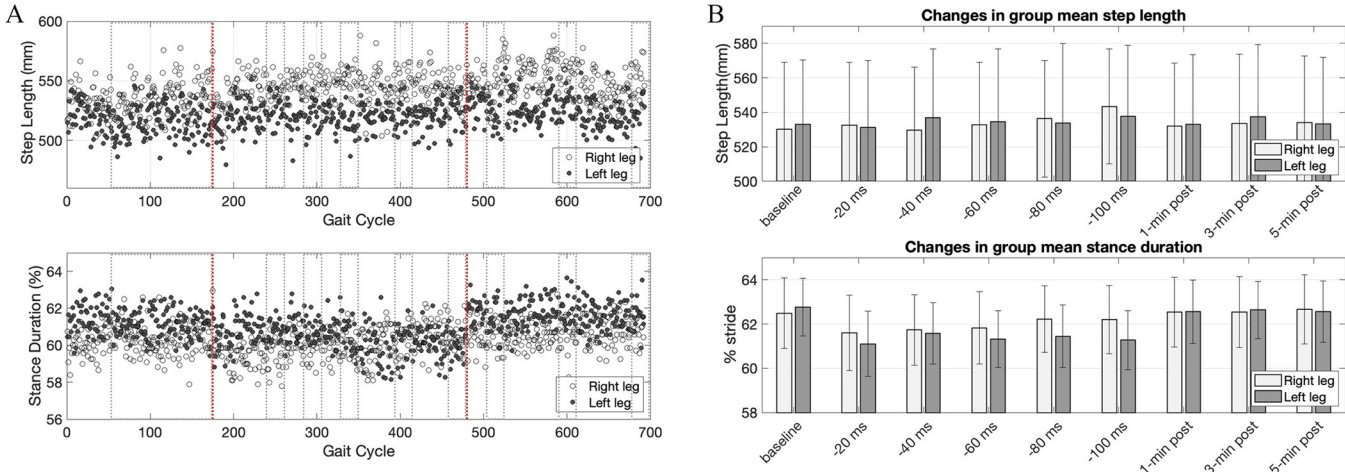

**Fig 2.** (**A**) Example of changes in step length and percent stance duration during an explicit trial conducted with a subject. The horizontal axis represents the gait cycle across a 16-minute trial, divided into baseline, perturbation (marked by two vertical lines), and post-perturbation periods. (**B**) Group data illustrating changes in step length and percent stance duration for explicit trials. The means were calculated from different epochs (highlighted by dotted boxes in Fig 2A): spanning over the final 3 minutes of the baseline and the 30-second interval for each perturbation level. For the post-perturbation period, the displayed values represent the means obtained from the 30-second periods within the 1-minute, 3-minute, and 5-minute durations following the completion of the FES perturbation.

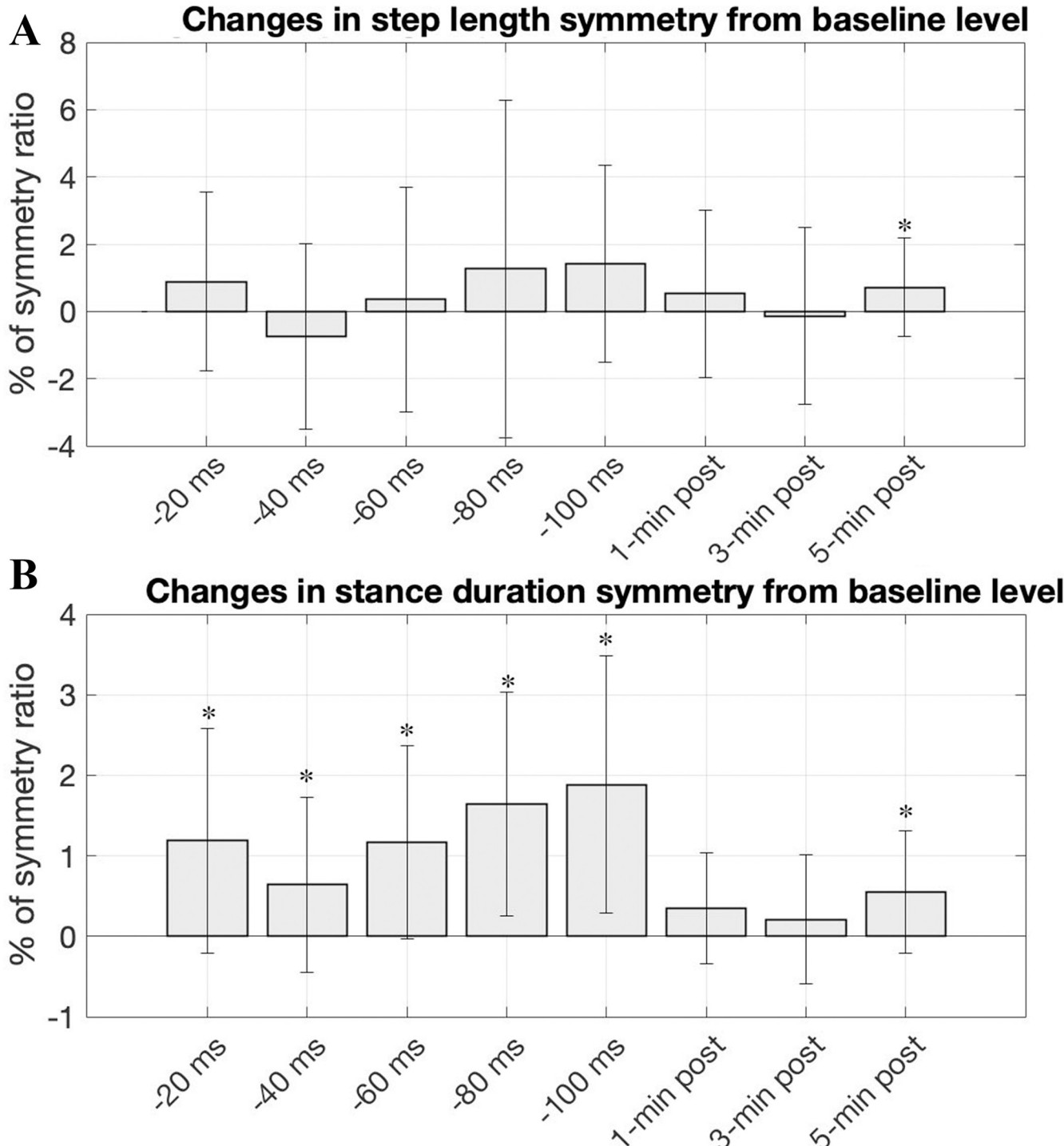

**Fig 3.** Explicit group results of step length symmetry (**A**) and average percent stance duration symmetry (**B**) averaged across 18 subjects over different epochs. Values equal to 0 represent perfect symmetry. The vertical lines show the standard deviation among subjects for each epoch. The asterisks (*) indicate where the induced symmetry values were shown to be significantly different from the baseline ($p < 0.05$).

induced during the perturbation period were statistically significant in comparison to the baseline value ($p = 0.0024$, $0.0305$, $0.0012$, $0.0003$, and $0.00032$, respectively for each perturbation epoch). During the post-perturbation period, the stance asymmetry induced by FES perturbation lingered slightly, but no statistical significance was observed except in the 5-min post-epoch ($p = 0.0149$). These alterations in stance duration asymmetry appeared to influence the spatial gait parameter (step length symmetry) as indicated by the larger variance in step length symmetry during the perturbation period (Fig 3A). Nevertheless, there was no consistent trend in the direction of step length asymmetry changes, nor were there any statistically significant changes in step length symmetry across different epochs except in the 5-min post-epoch ($p = 0.0469$). The statistical analysis conducting a one-way repeated measure ANOVA with post-hoc pairwise multiple comparisons, focusing solely on the perturbation period of the explicit trial, revealed that different perturbation levels significantly influenced the magnitude of altered stance symmetry ($F(4, 68) = 2.51$, $p \leq .05$). This result suggests a general trend of increasing stance duration asymmetry as perturbation levels increased.

To investigate how the subjects modified their gait patterns in response to the FES perturbation, we computed the percent mean time for each leg's heel strike and toe-off phases. We specifically examined the changes in the right stance and swing times compared to the left leg. Fig 4 illustrates the percent gait stance and swing time plot, with the mean percent time on the x-axis and different epochs vertically along the y-axis. During the perturbation period (Fig 4A), right toe-off and heel-strike times were shifted toward the left. Accordingly, the entire timeline for the right swing and stance was shifted to the left in response to FES perturbations. The shifting right swing and stance times decreased both right and left double support times compared to the baseline as shown in Fig 5A. In the post-perturbation period, the timeline shift seemed to revert toward the baseline, despite a minor residual effect.

Fig 5 shows the group mean changes in double limb support time for both legs under the explicit (A) and implicit (B) conditions. In the explicit trial, there was a notable decrease in double limb support time during the perturbation period. A statistically significant decrease in double limb support time was observed for both legs, yet no significant differences were detected between the right and left double limb support time (double limb support symmetry). Following the post-adaptation period, the double limb support time rapidly reverted to base levels. In the implicit trial, changes in double limb support time were minimal.

In the implicit trial, subjects did not make conscious actions in response to electrical stimulation. Fig 4B shows the percent gait stance and swing time from the implicit condition. Higher perturbation levels appeared to cause a slight leftward shift in the right swing and stance timeline, but no significant changes were observed. Fig 6A shows the changes in the group mean step length and group mean stance duration for the implicit trial. Throughout the perturbation period, a slight extension in the right stance was observed in juxtaposition to the left stance, but no statistical significance was found. Additionally, subjects did not exhibit any substantial changes in step length. Fig 6B shows the group mean of step symmetry changes and stance duration symmetry changes for the implicit condition. Concerning changes in the step length symmetry, no distinct trend was observed in the alterations of step length symmetry (Fig 6B top). Concerning changes in the stance symmetry, a slight trend toward positive changes in stance symmetry was observed during the perturbation period but not statistically significant, and this general trend vanished once the FES perturbation stopped. Statistical analysis conducted with a one-way repeated measures ANOVA revealed no significant effect of perturbation levels on changes in stance duration symmetry during the adaptation period. Similarly, no significant difference was observed during the post-adaptation period. In terms of step length symmetry, the variability of symmetry significantly increased during the perturbation period, and no substantial directional changes were observed.

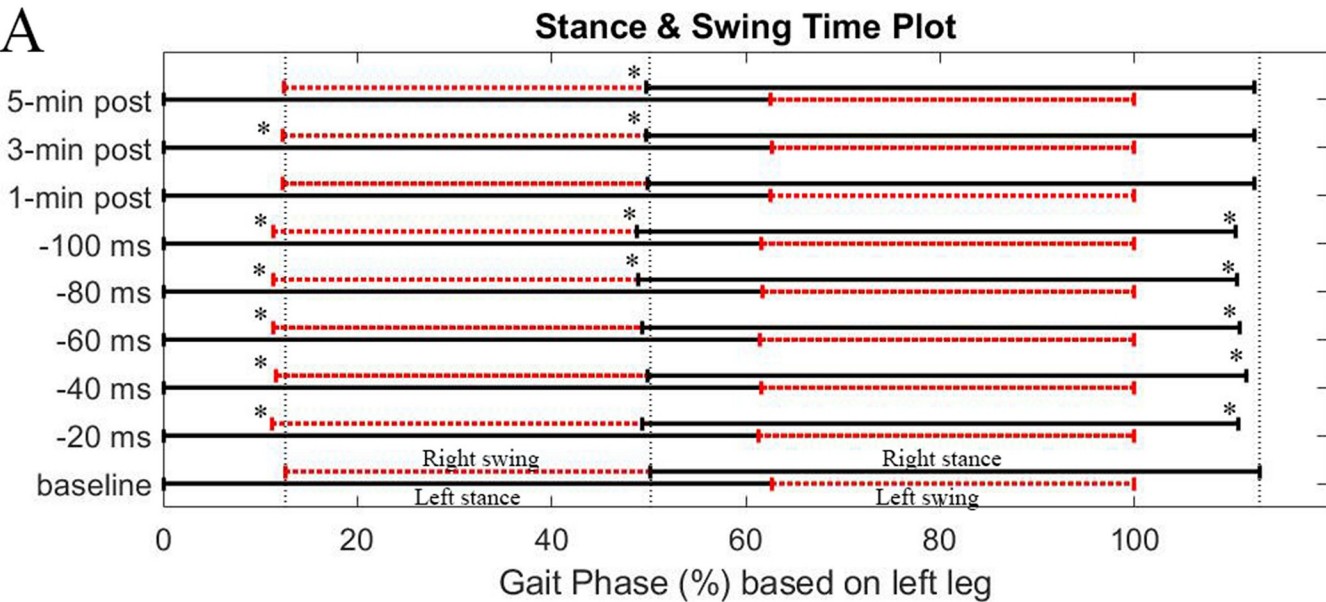

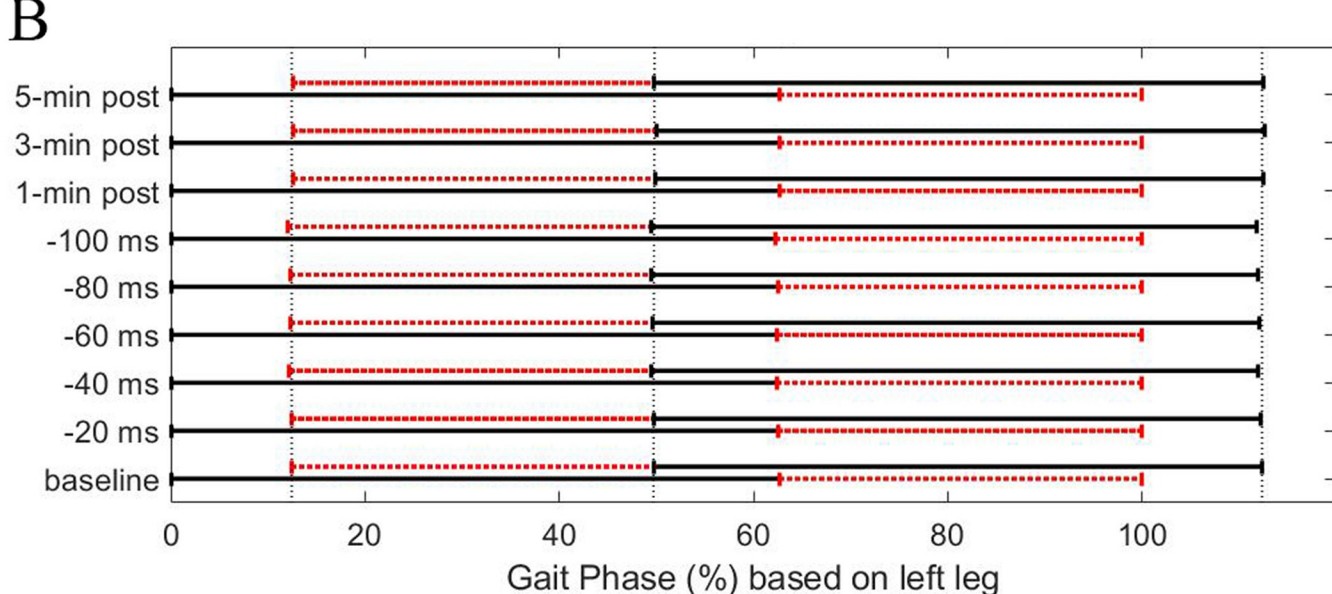

**Fig 4.** Changes in the percent duration of stance (represented by the solid line) and swing (represented by the dotted line), averaged across 18 subjects under both explicit (**A**) and implicit (**B**) conditions. Nine strides represent the stance and swing time obtained from each different epoch. In each epoch, the right stance and swing time are aligned with the left leg; the bottom and top lines represent the left and the right legs, respectively. The percentage times of the toe-off phase, heel strike, and the subsequent toe-off for the right leg were compared to those from the baseline. The asterisks (*) indicate statistically significantly significant differences from the baseline (p<0.05) for the phase timings.

## Discussion

In this study, we proposed a novel approach to utilizing FES as a perturbation method for altering symmetric gait patterns and evaluated the potential benefits of the FES perturbation by examining the extent of spatiotemporal gait asymmetry adaptation during treadmill walking. The achievement of gait symmetry is one of the primary goals in rehabilitation following neurological conditions. To address gait asymmetry, adaptive locomotor adjustments through motor training are essential. Rehabilitation devices, such as robotic gait orthosis or split-belt

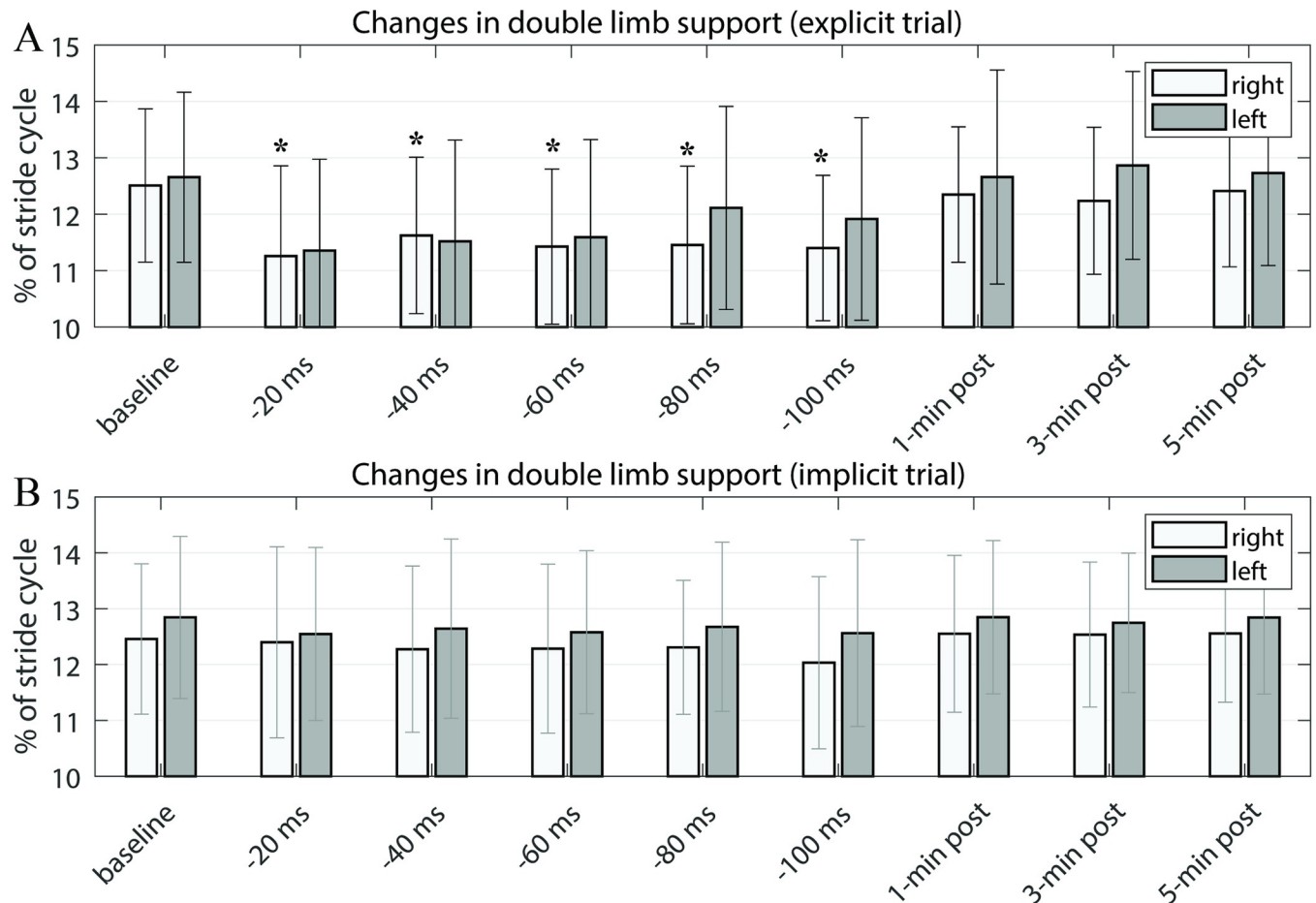

**Fig 5.** Changes in the percent double limb support time over different epochs, averaged across 18 subjects under both explicit (**A**) and implicit (**B**) conditions. The percentage time values for both legs were compared to those from the baseline. The asterisks (*) indicate statistically significant differences from the baseline (p<0.05).

treadmills, can facilitate locomotor adjustments. These mechanical devices not only assist subjects in walking but also impart perturbations to their gait. Robotic devices can produce viscous resistance to the leg [22], while split-belt treadmills can modify spatiotemporal gait patterns [11]. These perturbation-based training methods can prompt gait adaptation, which is maintained on a short-term basis, as demonstrated in aftereffects. Gait adaptation can also occur in response to non-mechanical perturbations, such as visual feedback perturbation and electrical stimulation [23, 24].

Our prior study tested the effects of external periodic perturbation using electrical stimulation on gait entrainment (the synchronization of gait cycle duration to the period of electrical stimulation) while walking on a treadmill. We discovered that subjects spontaneously synchronized their gait cycle duration to the period of the electrical stimulation when the stimulation periods were relatively close to the subject's initial gait period [16]. Remarkably, entrainment was still observed when the stimulation intensity was below the threshold that induced noticeable ankle movements. These observations suggest that periodic afferent signals evoked by FES may contribute to promoting certain features of entrainment, given that spinal neural networks significantly interact with afferent sensory feedback [25, 26]. In this study, we expanded upon this idea, utilizing FES as a perturbation method to induce gait asymmetry adaptation

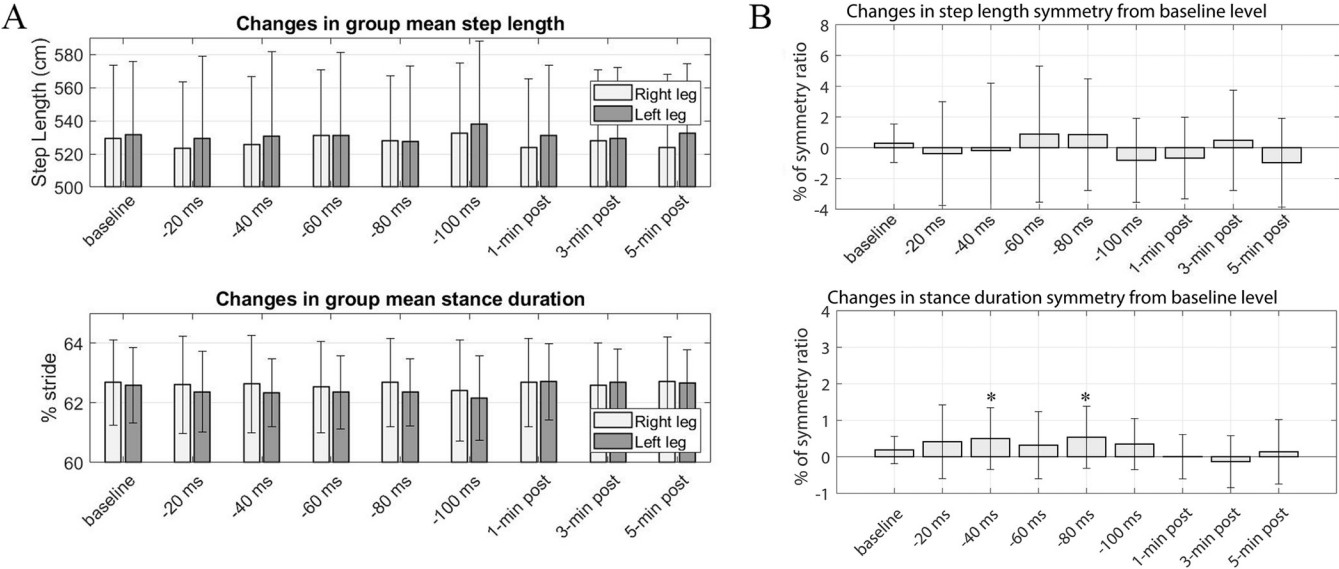

**Fig 6.** (**A**) Group data illustrating changes in step length and percent stance duration for implicit trials with vertical bars representing ±SD. Conventions are same as in Fig 2B. (**B**) Implicit group results of step length symmetry (*top*) and average percent stance duration symmetry (*bottom*) over different epochs, averaged across 18 subjects. The asterisks (*) indicate symmetry values that showed significant differences from the baseline measurement ($p<0.05$).

concerning spatiotemporal gait patterns. Perturbation-based training heavily relies on sensory inputs to provoke adaptive responses [27], with the option of using different types of perturbations. For example, visual or auditory cues can be employed as perturbations to engage corresponding sensory information in training. However, we believe that FES offers a simple and cost-effective way to introduce external perturbations during walking. FES not only disrupts gait patterns but also affects the somatosensory pathway, both directly and indirectly, during gait adaptation, leveraging brain plasticity to reshape the neural circuits involved in this process [14, 28]. In our FES perturbation procedure, we applied periodic electrical pulses to both legs, with a controllable time interval between the two pulses sent to each leg. Since no other studies have attempted to use FES in this manner toward inducing asymmetric gait adaptation, we measured the changes in step length and stance duration, and aimed to answer the following questions: (1) Does the use of electrical stimulations to both legs with an offset interval have an effect on alterations in spatiotemporal gait patterns (explicit trial)? and (2) If entrainment is partially a spontaneous phenomenon, does FES-induced perturbation still influence the alteration of spatiotemporal gait patterns even if a subject does not consciously respond to the perturbation (implicit trial)?

### Effects of FES-induced perturbations on temporal gait asymmetry adaptation (explicit trial)

In this study, we arranged the implicit trials to always precede the explicit trials to avoid any possible influence or recollection of subjects' explicit actions on their implicit trials. Here, we first analyzed the outcomes of the explicit trials to comprehend how subjects adapted their gait patterns in response to the FES perturbations. We focused on two adaptive gait parameters: step length and the percentage time in the stance phase. Step length is a spatial gait parameter while percent time in the stance phase (referred to as stance duration) is a temporal measure. During explicit trials, as subjects attempted to synchronize their right and left toe-off phase with the timing of the administered electrical pulse timings, their step length and stance

duration changed differently between the right and left leg (Fig 2). As the interval between the right electrical pulse and left electrical pulse decreased (targeting a shorter gap between the left toe-off and the subsequent right toe-off), we observed a decrease in the left stance relative to the right (Fig 2B bottom). Correspondingly, the left swing time became longer than the right swing time (not shown). This resulted in a positive directional change in stance duration symmetry (Fig 3B).

Initially, we thought that the right stance duration would decrease due to the altered timing interval between the two periodic electrical pulses, which was intended to slightly advance the phase of right-toe off. Contrary to our expectations, we observed that the subjects tended to adjust both their left and right gait steps in a complementary manner in response to the perturbation. This compensatory behavior resulted in reduced double limb support time for both legs (Fig 5A), accompanied by a slight decrease in the left stance duration (a corresponding subtle increase in the right stance duration) more than in the opposite leg. The precise mechanism behind why and how subjects adjusted their gait steps in this specific way is still being examined. Fig 3 shows overall changes in gait symmetry relative to the baseline level for the explicit trial (where subjects consciously attempted to entrain their gait with the perturbation). The stance duration symmetry tended toward a positive increase with higher perturbation levels (offset time). Overall, an increase in stance duration asymmetry was observed with higher perturbation levels, except the initial period when the perturbation started. The unexpectedly higher stance duration asymmetry at the lowest perturbation level may be due to subjects initially struggling to align their gait phase with the timing of the perturbations. The adapted stance duration asymmetry returned to symmetry once the perturbation was removed. However, subtle positive asymmetry persisted for up to 5 minutes in the post-perturbation period. These findings demonstrated that the FES perturbation procedure can modify subjects' temporal gait pattern without necessarily utilizing mechanical devices such as a split-belt treadmill (SBT), which requires physical interaction with the subjects' walking. Another important observation is the retention trend in the adapted temporal parameter. The positive directional stance duration asymmetry persisted slightly during the post-perturbation period, with only the last epoch showing statistical significance. While this may not provide conclusive evidence, this observation differs from studies using split-belt treadmills. In those studies, stance duration was rapidly changed during adaptation and both legs' stance time rapidly returned to baseline levels, as did stance time asymmetry [11]. This disparity may be due to differing environmental influences between mechanical-based and the FES-based perturbation approaches. FES perturbation did not involve speed asymmetry, while SBT includes physical speed changes between the right and left legs. Several combinations of other factors, such as psychological and biomechanical factors or afferent sensory signal actions, may influence adaptive behavior differently.

Although stride times for both legs were largely equivalent across all periods, the percentage time in stance, swing, and double limb support changed during the perturbation period. At baseline, stance, swing, and double support times were symmetric for both legs (Fig 4). During the perturbation period, the percent time of right toe-off and right heel-strike (start and end times of the right swing) were slightly advanced, leading to a slight leftward shift of the right swing and consequently right stance time as well relative to those of the left (Fig 4A). This change in right swing and stance times remained consistent throughout all perturbation levels. The shifting right swing and stance times decreased both right and left double support times compared to the baseline (Fig 5A). It was reported that SBT walking induces asymmetry in double limb support time, which is known to proportionally change with the split-belt speed ratio [29]. Although this study observed significant decrease in double limb support times for both legs (Fig 5A), it is important to highlight that no statistically significance differences were

found in the double limb support time symmetry during the perturbation period, in contrast to what is typically seen in SBT walking. This implies that physical speed differences between the right and left legs predominantly determine double limb support symmetry, which was not the case with our FES perturbation approach.

## Effects of FES-induced perturbations on spatial gait asymmetry adaptation (explicit trial)

Under the explicit condition, the FES perturbation procedure used in this study was primarily oriented toward altering stance and swing symmetry (temporal gait parameter) by entraining toe-off phase with stimulation timing. We anticipated that changes in temporal gait symmetry would inherently influence spatial gait symmetry, such as step length symmetry. However, the change in spatial gait pattern was not as significant as the observed temporal changes. A slight increase in the right step length was noticed at higher perturbation levels (Fig 2B top), but it was not statistically significant. Moreover, despite slight positive directional changes in step length symmetry during the perturbation period (Fig 3A), the large variance inhibited the determination of statistical significance. In summary, the response of step length changes to FES perturbation varied substantially across individuals. This implies that changes in temporal gait patterns are not necessarily tied to changes in spatial gait patterns. Essentially, accommodating temporal gait patterns does not obligate a modification of the spatial gait pattern, or vice versa. Other studies utilizing SBT demonstrated that interlimb coordination can be independently controlled and adjusted in the adult human nervous system without necessarily changing various aspects of intralimb coordination [29, 30]. This suggests that different gait parameters are maintained and adapted through separate processes. Our study's results are consistent with the prior studies' findings, indicating that temporal and spatial controls for symmetric gait can be adapted independently.

## Effects of FES perturbations under implicit condition

Effective interventions must facilitate efficient motor adaptation and ensure persistent changes (retention) in newly learned motor patterns. Various researchers have proposed that the retention benefit of learning relies on the unconscious or automatic mode within the learning process [19–21]. Therefore, we wanted to understand how FES perturbation affects gait asymmetry changes under implicit conditions. In the implicit trial, subjects walked on a treadmill while receiving electrical pulses but did not consciously alter their gait patterns. We observed a substantial variation in how subjects spontaneously compensated their gait in response to FES perturbation. When examining changes in step length and stance duration, we noticed a slightly reduced left stance duration compared to the right during the perturbation period (Fig 6A bottom), resulting in somewhat positive directional changes in stance duration symmetry (Fig 6B bottom). The degree of asymmetry was significantly smaller than that induced by the subjects' conscious actions (explicit trial). In the post-perturbation period, there were no signs of aftereffects of the asymmetric stance duration. Regarding the spatial gait pattern changes under the implicit condition, due to large variances across the subjects, we found no significant changes in step length or step length symmetry (Fig 6B top). Additionally, no substantial changes were observed in stance and double support time relative to either leg (Figs 5B and 6).

During implicit trials, we observed that some subjects were unwittingly subjected to perturbation, exhibiting little to no gait entrainment. Others showed substantial entrainment, but most showed only sporadic entrainment throughout the implicit trials. In our earlier study which utilized FES on only one leg, approximately 55% of subjects showed some degree of

entrainment [16]. However, the current study showed a lower proportion of subjects showing some degree of entrainment, which could be because FES perturbation was applied to both legs in this study, as opposed to one. It appears more challenging for both legs to spontaneously display entrainment to electrical pulses on each leg than to show entrainment to electrical pulses on a single leg.

We speculate that using a higher stimulation amplitude could have yielded more significant results. However, increasing the amplitude might have more robustly stimulated the muscles and led to greater ankle torque, which was not our intent as a method of inducing gait asymmetry. We maintained minimal stimulation amplitude throughout the implicit trials, as we aimed to understand the extent to which sensory afferent signals caused by FES influence symmetric gait adaptation. If the stimulation intensity needs to be increased to induce significant symmetric gait adaptation spontaneously, this inevitably results in uncomfortable tactile sensations and necessitates incorporation of electrically evoked joint torque as an additional factor in gait adaptation. Another reason for the lack of distinct results under the implicit condition could be that even when entrainment occurred spontaneously in walking, it did not necessarily synchronize with toe-off phases. Some subjects showed entrainment to their heel-strike phases. Consequently, the outcomes exhibited by subjects varied, leading to greater variations in spatiotemporal gait symmetry changes. Under implicit conditions, an experimental protocol designed to accommodate individual differences would likely yield more significant results. Therefore, this suggests the need for further investigation in this area.

## Limitations and further direction

The current study represents a pioneering effort to evaluate a novel FES-based perturbation procedure to induce gait asymmetry adaptation during treadmill walking. There are, however, several limitations to consider. First, the stimulation amplitude utilized in this study was confined to a minimum threshold; therefore, the results might be different if varied stimulation intensities were used. For instance, as the stimulation amplitude increases, stronger joint torque follows, which may contribute to inducing asymmetric gait adaptations differently through FES-evoked sensory signals involved in gait control via neural circuits in the spinal cord. Second, the maximum offset time for the perturbation used in this study was capped at 100 ms, which is roughly equivalent to 7.5% of the average gait cycle duration. Should the perturbation become larger, entrainment is unlikely to occur during treadmill walking due to the constraints on walking speed, which is fixed on a treadmill, putting boundaries on gait entrainment. If experiments can be conducted in an over-ground walking setting, the perturbation size could be adjusted more freely, making the effects of FES perturbation more visible. Additionally, due to variations in walking speed and gait styles among the subjects, the range of the offset time ($\Delta\tau$) did not correspond to the exact percentage time of the gait cycle duration. If gait favorably entrains to a particular percentage of the gait cycle duration, then the offset time should be adjusted based on an exact percentage time of each individual's gait cycle duration, rather than using an absolute time for more significant results. Third, this study did not explore how subjects' inherent baseline inherent asymmetry tendencies may affect the directional changes in gait symmetries in response to FES. Although the baseline asymmetry's statistical significance is negligible, it would be valuable to investigate this further. Another limitation is the short post-FES perturbation observation period, lasting only 5 minutes. To explore the long-term effect, an extended post-perturbation period should be examined to accurately assess any retention effects. These limitations highlight the necessity for further research.

## Conclusion

This study investigated the efficacy of FES perturbation-based gait training, which employs periodic electrical pulses to both legs, to cultivate effective gait adaptation in spatial and temporal gait parameters: step length and stance duration. We found that FES perturbation training, where subjects consciously synchronized their gait to the stimulation, induced noticeable asymmetry in stance time that persisted marginally over a 5-minute post-perturbation period. Although not statistically significant, a slight trend of changes in stance time symmetry by FES was observed even without conscious effort, possibly suggesting that the changes in stance time asymmetry include spontaneous components to some extent. The changes in temporal gait symmetry did not correspond well with the changes in step length symmetry. This indicates that temporal and spatial controls for symmetric gait can adapt separately. It is our hope that electrical stimulation can serve as an alternate cost-effective method for gait rehabilitation, helping patients improve their gait asymmetry efficiently by facilitating gait asymmetry adaptation processes during treadmill walking training. Moreover, electrical stimulation has shown potential to enhance muscle strength and stability [31]. Hence, incorporating entrainment through FES may be advantageous in aiding individuals with gait impairments. Comprehensive future research measuring the degree of interaction between the explicit and implicit control of gait is necessary to fully understand the mechanisms at play in the FES-based perturbation approach.

## Acknowledgments

Authors would like to thank the support provided by the College of Engineering at California Baptist University.

## Author Contributions

**Conceptualization:** Seung-Jae Kim.

**Investigation:** Seung-Jae Kim, Alex Worthy, Brandom Lee, Setareh Jafari, Olivia Dyke, Jeonghee Cho.

**Writing – original draft:** Seung-Jae Kim.

**Writing – review & editing:** Elijah Brown.

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
