## [Decision Letter · Decision Letter 0]

26 Aug 2024

PONE-D-24-11792Adapting spatiotemporal gait symmetry to functional electrical stimulation during treadmill walkingPLOS ONE

Dear Dr. Kim,

Thank you for submitting your manuscript to PLOS ONE. After careful consideration, we feel that it has merit but does not fully meet PLOS ONE’s publication criteria as it currently stands. Therefore, we invite you to submit a revised version of the manuscript that addresses the points raised during the review process.

We look forward to receiving your revised manuscript.

Kind regards,

Kei Masani

Academic Editor

PLOS ONE

Journal Requirements:

Additional Editor Comments:

I am sorry that the process took longer than usual. 

Both reviewers agreed the paper has merits while requiring clarification in the presentation. Please follow the reviewers' comments to improve readability.

Reviewers' comments:

Reviewer's Responses to Questions

**Comments to the Author**

1. Is the manuscript technically sound, and do the data support the conclusions?

Reviewer #1: Partly

Reviewer #2: Partly

2. Has the statistical analysis been performed appropriately and rigorously? 

Reviewer #1: Yes

Reviewer #2: No

3. Have the authors made all data underlying the findings in their manuscript fully available?

Reviewer #1: Yes

Reviewer #2: Yes

4. Is the manuscript presented in an intelligible fashion and written in standard English?

Reviewer #1: Yes

Reviewer #2: Yes

5. Review Comments to the Author

Reviewer #1: General Comments

(1) In general, the gait cycle varies among different subjects, so there should be some variability in gait cycle duration and the rate of stance and swing phases among subjects. In your experimental setup, the offset time of FES perturbation is determined as the absolute time from 0 to -100 ms. If there is some difference in gait cycle duration among subjects, it seems the perturbation timing is applied at different phases for each subject. What percentage is the Δτ of each subject in the gait cycle? Please indicate and discuss the influence of inter-subject variability on actual gait cycle duration and the rate of stance and swing phases.

(2) There is usually a slight gait asymmetry in normal walking, even in young healthy subjects, referred to as the “baseline” in this paper. In fact, in Fig. 2A, a typical subject’s baseline results also seem slightly asymmetrical; the right step length is larger than the left one, and the right stance duration is shorter than the left one, although the statistical significance of this difference is not clear. Such tendencies of gait asymmetry in normal walking may vary among subjects, with some showing the opposite tendency. Please indicate and discuss the influence of baseline asymmetry tendencies of each subject on the main perturbation results.

Specific Comments

(1) In Fig. 2A, the step length scale might be incorrect: (cm) → (mm)?

Reviewer #2: This study by Kim and colleagues investigated whether asymmetric FES could induce asymmetric walking in healthy young adults. The study is innovative and generally robust in their methodology and I commend the authors for that. On the other hand, the interpretation and reporting of the findings could still be refined. Kindly find specific comments below:

Abstract

Line 23: Is the aim of the study to improve or disrupt gait symmetry?

Line 29: Perhaps helpful to mention that the explicit trial was performed in another session.

Line 33: It is important to differentiate between results that are statistically significant and those that are not. The implicit condition failed to consistently induce temporal or spatial adaptation, even with large perturbations, and this should be clearly acknowledged.

Line 34: The right stance became relatively longer. If I look at Figure 2B, it seems like the most significant change in stance duration from baseline, is reduction in the left leg rather than increase in the right leg. The interpretation should be refined and adjusted throughout.

Line 38: Implications should be focused on the explicit response, as the implicit response was rather weak and inconsistent.

Introduction

The focus of the previous study was implicit adaptation to the FES, and here you introduce implicit adaptation to be relevant for rehabilitation. However the subsequent result and discussion section heavily focuses on explicit adaptation, and this seems confusing. Could you introduce why explicit adaptation is given this importance, and what the advantages of using FES for explicit adaptation compared to visual or auditory cues might be?

Methods

Kindly mention that the treadmill was a split-belt treadmill.

FES timing: How functional was the FES? Unclear why the FES was applied at toe-off when the plantar flexors already start working from mid- to terminal stance. Toe-off is when the hip and ankle flexors take over to accelerate the leg through swing phase.

FES – gait synchronization: Once the time period was calculated from the toe-offs, were the toe-offs still used to determine when each stimulation would be given or did the stimulation just occur at a preset rate irrespective of whether the left leg was in toe-off or not? Kindly clarify.

Heel-strike determination: Often heel-strikes occur after the foot achieves its most forward position (which occurs in the air, before putting the foot down on the treadmill). This could impact the parameters of participants who walk with this pattern. Why not use the force plates from the treadmill, or the change in velocity of the markers to determine heel-strikes?

Statistics – were all the outcomes normally distributed and did the models fit the data well?

Statistics Line 202 – Typically, post-hoc t-tests are only performed if there is a significant effect in the repeated measures ANOVA. Otherwise you run into multiple comparison problems.

Results

Figures 3 (Explicit) and 6 (Implict) Asymmetry – the scales on the axis of the figures are different. Kindly harmonize.

Results are rather descriptive – would prefer to start with inferential statistics and then describe those that are significant. Further, there is no need to repeat the description of the figures in the text and in the legend of the figure.

If you want to show a successive increase in asymmetry with larger perturbations, you would need to check the t-tests for pairs of perturbation levels, not by comparing with baseline. And correct for multiple comparisons!

Discussion

The differences in asymmetry between perturbation levels are not discussed. Both for explicit and implicit adaptation, some unexpected responses are seen at low or high perturbation levels.

The reason why post-adaptation changes were only significant in the last 30 seconds is not clear. Were there noticeable changes in their gait pattern following the FES? What was the subjective perception after removing the FES?

What was the amount of perturbation in relation to the baseline time period between heel strikes? Generally, we refer to perturbations as a ratio to baseline parameters – if you can provide information about the size of the ratio between the legs, that might help to interpret how small or large the perturbation was in relation to other studies.

6. PLOS authors have the option to publish the peer review history of their article (what does this mean?). If published, this will include your full peer review and any attached files.

Reviewer #1: No

Reviewer #2: **Yes: **Nicholas D'Cruz

---

## [Author Response · Author response to Decision Letter 0]

1 Sep 2024

Response to Reviewer's Comments

We would like to thank the reviewers for their valuable comments. The reviewers’ comments are listed below in black, and the authors’ response is blue.

Reviewer #1: General Comments

(1) In general, the gait cycle varies among different subjects, so there should be some variability in gait cycle duration and the rate of stance and swing phases among subjects. In your experimental setup, the offset time of FES perturbation is determined as the absolute time from 0 to -100 ms. If there is some difference in gait cycle duration among subjects, it seems the perturbation timing is applied at different phases for each subject. What percentage is the Δτ of each subject in the gait cycle? Please indicate and discuss the influence of inter-subject variability on actual gait cycle duration and the rate of stance and swing phases.

>>We agreed that the reviewer raised an important point. The gait cycle duration changes with walking speed, so we did not quantitatively analyze the variability in gait cycle duration across all the subjects. However, based on our measurements, the average gait cycle duration was 1339 ± 40 ms, and the offset ∆τ of 0~100 mm then represents roughly 0 to 7.5% of the gait cycle duration. Although the 20 ms offset time increment (roughly 1.5% of gait cycle duration) doesn’t correspond to the same fraction (%) of the gait cycle duration among subjects, the changes in gait asymmetric patterns occurred in response to FES because the subjects synchronized their gait phase with the stimulation timing. In a future study with a more rigorous protocol, we could apply the offset time as a percentage of each individual’s gait cycle duration. We have added this concern as an additional limitation in the Discussion section, which now reads: 

“… due to variations in walking speed and gait styles among the subjects, the range of the offset time (∆τ) did not correspond to the exact percentage time of the gait cycle duration across all the subjects. If gait favorably entrains to a particular percentage of the gait cycle duration, then the offset time should be adjusted based on a exact percentage time of each individual’s gait cycle duration, rather than using an absolute time for more significant results”

(2) There is usually a slight gait asymmetry in normal walking, even in young healthy subjects, referred to as the “baseline” in this paper. In fact, in Fig. 2A, a typical subject’s baseline results also seem slightly asymmetrical; the right step length is larger than the left one, and the right stance duration is shorter than the left one, although the statistical significance of this difference is not clear. Such tendencies of gait asymmetry in normal walking may vary among subjects, with some showing the opposite tendency. Please indicate and discuss the influence of baseline asymmetry tendencies of each subject on the main perturbation results.

>>We thank the reviewer for raising this question. Although it would be interesting to quantitatively examine how an individual’s baseline asymmetry tendency affects the directional tendency of changes in gait symmetry in response to FES, we do not believe we would find any significant trends. Our previous experiences indicate that baseline asymmetry tendencies are not consistent even for the same subject (for example, the direction of baseline asymmetry is not consistent for different trials). However, we do not rule out the possible effects of baseline asymmetry on the main results, and a more rigorous study could address this. We have added this point in the Discussion section, which now reads: 

“… this study did not explore how subjects’ inherent baseline inherent asymmetry tendencies might affect the directional changes in gait symmetries in response to FES. Although the baseline asymmetry’s statistical significance is negligible, it would be valuable to investigate this further.”

Specific Comments

(1) In Fig. 2A, the step length scale might be incorrect: (cm) → (mm)?

>>We thank the reviewer. We have corrected the typo.

Reviewer #2: This study by Kim and colleagues investigated whether asymmetric FES could induce asymmetric walking in healthy young adults. The study is innovative and generally robust in their methodology and I commend the authors for that. On the other hand, the interpretation and reporting of the findings could still be refined. Kindly find specific comments below:

Abstract

Line 23: Is the aim of the study to improve or disrupt gait symmetry?

>>The study aimed to explore the potential of using FES as a perturbation method during walking to promote gait asymmetry adaptation, with the long-term goal of improving gait asymmetry. We have revised the sentence to clarify the objective, which now reads: 

“This study explored the potential of using functional electrical stimulation (FES) as a perturbation method during treadmill walking to promote gait symmetry adaptation by investigating whether the FES perturbation paradigm could induce gait symmetry adaptation concerning spatial and temporal gait patterns in healthy subjects”

Line 29: Perhaps helpful to mention that the explicit trial was performed in another session.

>>We have revised the sentence by clearly mentioning two trials (implicit and explicit), which now reads: 

“Subjects participated in two trials: implicit and explicit. In the implicit trial, they walked comfortably during FES perturbation without consciously adjusting their gait. In the explicit trial, they voluntarily synchronized their toe-off phase to the stimulation timing.”

Line 33: It is important to differentiate between results that are statistically significant and those that are not. The implicit condition failed to consistently induce temporal or spatial adaptation, even with large perturbations, and this should be clearly acknowledged.

>>Although the observed changes in temporal gait symmetry in the implicit trial were not statistically significant over the entire varying perturbation levels (-20 ~ -100 ms), there were partial significant differences compared to the baseline symmetry (Fig. 6B bottom). We are sorry for the confusion, and we agree with the reviewer that the results should be clearly and accurately described in the Abstract. Accordingly, we have revised the sentence, which now reads:

“In this study, during the explicit trial, subjects adapted their gait patterns to the electrical pulses, resulting in a directional change in stance time (temporal) symmetry, with the left stance becoming shorter than the right. The stance time asymmetry induced by FES perturbation showed a slight residual effect. In the implicit trial, the directional change trend was slightly observed but not statistically significant.” 

Line 34: The right stance became relatively longer. If I look at Figure 2B, it seems like the most significant change in stance duration from baseline, is reduction in the left leg rather than increase in the right leg. The interpretation should be refined and adjusted throughout.

>>We thank the reviewer for bringing up an interesting point. Our original intention was to make the right-toe-off phase occur slightly earlier than usual, so we naturally focused on interpreting the observed change relative to the right side in comparison to the other side. We have revised our interpretation as suggested by the reviewer, throughout the manuscript. 

Line 38: Implications should be focused on the explicit response, as the implicit response was rather weak and inconsistent.

>>We have revised the sentence by adding the following statement:

“The implicit condition showed a similar slight trend but was not consistently statistically significant”

Introduction:

The focus of the previous study was implicit adaptation to the FES, and here you introduce implicit adaptation to be relevant for rehabilitation. However the subsequent result and discussion section heavily focuses on explicit adaptation, and this seems confusing. Could you introduce why explicit adaptation is given this importance, and what the advantages of using FES for explicit adaptation compared to visual or auditory cues might be?

>>We appreciate the valuable comments from the reviewer. Initially, we anticipated significant changes in gait symmetries under both explicit and implicit conditions. However, our experimental results showed minimal significant changes in the implicit conditions for various reasons. Instead of reporting only the explicit condition results, we decided to include findings from both conditions, as we believe it is still valuable to document the weak cases given that our study explored a novel method for inducing gait symmetry adaptation through FES. 

Following the reviewer’s suggestion, we have revised the Introduction to include statements about the potential use FES for explicit adaptation, etc. The revised text now reads:

“… We investigated whether such a perturbation paradigm utilizing FES could induce changes in spatial and temporal gait patterns as subjects voluntarily synchronized a certain gait phase of both legs with the perturbations (explicit condition). Motor learning includes a form of learning process known as use-dependent plasticity, which involves developing movement biases that favor repeated movement directions (17)…”

>> We have included a statement in the Discussion section that outlines the potential advantages of using FES over visual or auditory cues. The added text now reads:

“Perturbation-based training heavily relies on sensory inputs to provoke adaptive responses (27), with the option of using different types of perturbations. For example, visual or auditory cues can be employed as perturbations to engage corresponding sensory information in training. However, we believe that FES offers a simple and cost-effective way to introduce external perturbations during walking. FES not only disrupts gait patterns but also affects the somatosensory pathway, both directly and indirectly, during gait adaptation, leveraging brain plasticity to reshape the neural circuits involved in this process (14,28).”

Methods:

Kindly mention that the treadmill was a split-belt treadmill.

>>Although this study used a split-belt treadmill, both belts operated at the same speed, functioning like a traditional treadmill. Therefore, we chose not to mention the use of a split-belt treadmill to avoid any potential confusion.

FES timing: How functional was the FES? Unclear why the FES was applied at toe-off when the plantar flexors already start working from mid- to terminal stance. Toe-off is when the hip and ankle flexors take over to accelerate the leg through swing phase.

>> The intensity of FES was mild, resulting in only slightly noticeable ankle movement and a tingling sensation. During the explicit trial, subjects were asked to voluntarily synchronize their toe-off phase with the stimulation, as the slight ankle plantarflexion induced by FES was expected to resemble the ankle motion at toe-off. However, there might be some variability in the alignment of the gait phase with the stimulation, with some instances occurring slightly before or after the toe-off phase.

FES – gait synchronization: Once the time period was calculated from the toe-offs, were the toe-offs still used to determine when each stimulation would be given or did the stimulation just occur at a preset rate irrespective of whether the left leg was in toe-off or not? Kindly clarify.

>> The stimulation occurred at a present rate (i.e., at the same interval as the initial stride duration) irrespective of whether the left leg was in toe-off or not. We have provided additional clarification and revised the sentence, which now reads:

“During the perturbation period, we administered electrical pulses (0.12-second duration, 200 µs biphasic pulses at 90 Hz, 20-25 mA intensity) to both the right and left legs, at the same period of the baseline stride duration. In other words, periodic electrical stimulation was applied to each leg independently, but the stimulations were not delivered simultaneously to both legs; there was a time interval of τ between the stimulations.”

Heel-strike determination: Often heel-strikes occur after the foot achieves its most forward position (which occurs in the air, before putting the foot down on the treadmill). This could impact the parameters of participants who walk with this pattern. Why not use the force plates from the treadmill, or the change in velocity of the markers to determine heel-strikes?

>> In this study, we focused on analyzing gait symmetry between the right and left measurements. Although the exact timing of heel-strike may slightly differ between our measurement and others, such as those taken with a force plate, we believe that the symmetry measurement will remain unaffected. We were not able to use force plate, as the instrumental treadmill used in this study was not equipped with force plate. 

Statistics – were all the outcomes normally distributed and did the models fit the data well?

>>We tested for normality using the Kolmogorov-Smirnov method (SPSS), and the null hypothesis was not rejected, indicating that the data are approximately normally distributed. 

The reviewer may have a different opinion, but regardless of the normality test results, we believe that non-parametric statistical methods are not deemed necessary in this study. Although our sample size is not large enough, it falls within the typical range commonly used in similar studies investigating gait adaptation. Also, the decision against non-parametric analysis is supported by the understanding that non-parametric methods come with their own set of assumptions. 

Statistics Line 202 – Typically, post-hoc t-tests are only performed if there is a significant effect in the repeated measures ANOVA. Otherwise you run into multiple comparison problems.

>>We agreed with the reviewer on this point and appreciate this feedback. We revisited our statistical analysis and conducted pairwise comparisons following the ANOVA. The manuscript has been revised accordingly in the Result section. 

Results:

Figures 3 (Explicit) and 6 (Implicit) Asymmetry – the scales on the axis of the figures are different. Kindly harmonize.

>>We have revised Figure 6B to match the scale of figure 3. 

Results are rather descriptive – would prefer to start with inferential statistics and then describe those that are significant. Further, there is no need to repeat the description of the figures in the text and in the legend of the figure.

>>We appreciate the reviewer’s suggestion. We have edited the Results section to make it clearer and more concise by deleting redundant sentences. 

If you want to show a successive increase in asymmetry with larger perturbations, you would need to check the t-tests for pairs of perturbation levels, not by comparing with baseline. And correct for multiple comparisons!

>>We understand and appreciate the reviewer’s feedback. For our statistical analysis using ANOVA, we included only the data observed during the perturbation, excluding the baseline data, as our focus was to determine how increasing perturbation levels affect changes in symmetry magnitude. Therefore, we retained the paired t-test analysis comparing with the baseline, primarily to emphasize that the asymmetry observed in the explicit trail resulted from the perturbation.

Discussion:

The differences in asymmetry between perturbation levels are not discussed. Both for explicit and implicit adaptation, some unexpected responses are seen at low or high perturbation levels.

>> The unexpected responses observed at the lowest perturbation level in the explicit condition might be due to subjects initially struggling to align their gait phase with the timing of perturbations. On the other hand, the unexpected responses at the highest perturbation in the implicit condition might result from subjects’ gait phase not being consistently entrained during the perturbation level. We have added a sentence describing the tendency of increasing stance duration symmetry with higher perturbation levels (explicit trial only), which now reads:

“Overall, an increase in stance duration as

---

## [Decision Letter · Decision Letter 1]

25 Sep 2024

PONE-D-24-11792R1Adapting spatiotemporal gait symmetry to functional electrical stimulation during treadmill walkingPLOS ONE

Dear Dr. Kim,

Thank you for submitting your manuscript to PLOS ONE. After careful consideration, we feel that it has merit but does not fully meet PLOS ONE’s publication criteria as it currently stands. Therefore, we invite you to submit a revised version of the manuscript that addresses the points raised during the review process.

Please address the minor edits suggested by Reviewer 2 to further enhance the clarity of your manuscript.

We look forward to receiving your revised manuscript.

Kind regards,

Kei Masani

Academic Editor

PLOS ONE

Journal Requirements:

Additional Editor Comments:

Please address the minor edits suggested by Reviewer 2 to further enhance the clarity of your manuscript.

Reviewers' comments:

Reviewer's Responses to Questions

**Comments to the Author**

1. If the authors have adequately addressed your comments raised in a previous round of review and you feel that this manuscript is now acceptable for publication, you may indicate that here to bypass the “Comments to the Author” section, enter your conflict of interest statement in the “Confidential to Editor” section, and submit your "Accept" recommendation.

Reviewer #1: All comments have been addressed

Reviewer #2: All comments have been addressed

2. Is the manuscript technically sound, and do the data support the conclusions?

Reviewer #1: Yes

Reviewer #2: Yes

3. Has the statistical analysis been performed appropriately and rigorously? 

Reviewer #1: Yes

Reviewer #2: Yes

4. Have the authors made all data underlying the findings in their manuscript fully available?

Reviewer #1: Yes

Reviewer #2: Yes

5. Is the manuscript presented in an intelligible fashion and written in standard English?

Reviewer #1: Yes

Reviewer #2: Yes

6. Review Comments to the Author

Reviewer #1: (No Response)

Reviewer #2: Thanks for taking on the feedback positively and for adapting the manuscript!

Some small things - I noticed that the abstract in the manuscript submission system is not adapted to reflect the changes made. There is also still some incongruency between the first paragraph of the discussion and the abstract, in regard to whether the perturbation induces gait symmetry or asymmetry adaptations. Apart from these textual edits, I have no concerns that should keep this from being published.

Congratulations and all the best!

7. PLOS authors have the option to publish the peer review history of their article (what does this mean?). If published, this will include your full peer review and any attached files.

Reviewer #1: No

Reviewer #2: **Yes: **Nicholas D'Cruz

---

## [Author Response · Author response to Decision Letter 1]

27 Sep 2024

Response to Reviewer's Comments

We would like to thank the reviewers for their valuable comments. 

Reviewer #1: Reviewer #2: Thanks for taking on the feedback positively and for adapting the manuscript!

Some small things - I noticed that the abstract in the manuscript submission system is not adapted to reflect the changes made. 

>> Thank you for the information. Honestly, we're unsure why the changes we made didn't reflect in the revised PDF. We hope that this time, the submission system will correctly update the abstract with the revisions.

There is also still some incongruency between the first paragraph of the discussion and the abstract, in regard to whether the perturbation induces gait symmetry or asymmetry adaptations. Apart from these textual edits, I have no concerns that should keep this from being published.

>> We aimed to induce gait asymmetry during training through FES perturbation, ultimately facilitating the adaptation of asymmetric gait patterns. We have revised the first sentence of the Discussion, which now reads:

“In this study, we proposed a novel approach to utilizing FES as a perturbation method for altering symmetric gait patterns and evaluated the potential benefits of the FES perturbation by examining the extent of spatiotemporal gait asymmetry adaptation during treadmill walking.”

Congratulations and all the best!

>> Thank you for taking the time to review our manuscript. We greatly appreciate your feedback, which has helped us make significant improvements.

---

## [Editor Report · Decision Letter 2]

4 Oct 2024

Adapting spatiotemporal gait symmetry to functional electrical stimulation during treadmill walking

PONE-D-24-11792R2

Dear Dr. Kim,

We’re pleased to inform you that your manuscript has been judged scientifically suitable for publication and will be formally accepted for publication once it meets all outstanding technical requirements.

Kind regards,

Kei Masani

Academic Editor

PLOS ONE
---

## [Editor Report · Acceptance letter]

8 Oct 2024

PONE-D-24-11792R2 

PLOS ONE

Dear Dr. Kim, 

I'm pleased to inform you that your manuscript has been deemed suitable for publication in PLOS ONE. Congratulations! Your manuscript is now being handed over to our production team.

Kind regards, 

on behalf of

Dr. Kei Masani 

Academic Editor

PLOS ONE